



# The C$_{32}$ alkane-1,15-diol as a proxy of late Quaternary riverine input in coastal margins

Julie Lattaud[1*], Denise Dorhout[1], Hartmut Schulz[2], Isla S. Castañeda[1,4], Jaap S. Sinninghe

Damsté[1,3], Stefan Schouten[1,3]

*[1]NIOZ Royal Netherlands Institute for Sea Research, Department of Marine Microbiology and*

*Biogeochemistry, and Utrecht University, The Netherlands*

*[2] University of Tübingen, Department of Geosciences, Hölderlinstraasse 12, D-72074 Tübingen,*

*Germany*

*[3]Utrecht University, Department of Earth Sciences, Faculty of Geosciences, Budapestlaan 4, 3584*

*CD Utrecht, The Netherlands.*

*[4] Present address: University of Massachusetts, Department of Geological sciences, 244 Morrill*

*Science Center, Amherst, United States of America*

*\*Corresponding author: Julie.lattaud@nioz.nl*





ABSTRACT
The study of past sedimentary records from coastal margins allows us to reconstruct variations
of terrestrial input into the marine realm and to gain insight into continental climatic variability.
There are numerous organic proxies for tracing terrestrial input into marine environments but
none that strictly reflect riverine organic matter input. Here, we test the fractional abundance of
the $C_{32}$ alkane 1,15-diol relative to all 1,13- and 1,15-diols ($F_{1,15-C_{32}}$) as a tracer of riverine input
in the marine realm in surface and Quaternary (0-45 ka) sediments on the shelf off the Zambezi
and nearby smaller rivers in the Mozambique Channel (western Indian Ocean). A Quaternary (0-
22 ka) sediment record off the Nile River mouth in the Eastern Mediterranean was also studied
for diols. For the Mozambique Channel, surface sediments of sites most proximal to
Mozambique rivers showed the highest $F_{1,15-C_{32}}$ (up to 10%). The sedimentary record shows high
(15-35%) pre-Holocene $F_{1,15-C_{32}}$ and low (<10%) Holocene $F_{1,15-C_{32}}$ values, with a major decrease
between 18 and 12 ka. $F_{1,15-C_{32}}$ is significantly correlated ($r^2$=0.83, p<0.001) with the BIT index, a
proxy for soil and riverine input, which declines from 0.25-0.60 for the pre-Holocene to <0.10
for the Holocene. This decrease of both $F_{1,15-C_{32}}$ and the BIT is interpreted to be mainly due to an
increasing sea level, which caused the Zambezi River mouth to become more distal to our study
site, thereby decreasing riverine input at the core location. Some small discrepancies are
observed between the records of the BIT index and $F_{1,15-C_{32}}$ for Heinrich Event 1 (H1) and
Younger Dryas (YD), which can be explained by a change in soil sources in the catchment area
rather than a change in river influx. Like for the Mozambique Channel, a significant correlation
between $F_{1,15-C_{32}}$ and the BIT index ($r^2$=0.38, p<0.001) is observed for Eastern Mediterranean
Nile record.  Here also, the BIT index and $F_{1,15-C_{32}}$ are lower in the Holocene than in the pre-



Holocene, which is likely due to the sea level rise. In general, the differences between BIT index
and $F_{1,15-C32}$ Eastern Mediterranean Nile records can be explained by the fact that the BIT index
is not only affected by riverine runoff but also by vegetation cover with increasing cover leading
to lower soil erosion. Our results confirm that $F_{1,15-C32}$ is a complementary proxy for tracing
riverine input of organic matter into marine shelf settings and, in comparison with other
proxies, it seems not to be affected by soil and vegetation changes in the catchment area.



## 1.Introduction

Freshwater discharge from river basins into the ocean has an important influence on the
dynamics of many coastal regions. Terrestrial organic matter (OM) input by fluvial and aeolian
transport represents a large source of OM to the ocean (Schlesinger and Melack, 1981). Deltaic
and marine sediments close to the outflow of large rivers form a sink of terrestrial OM and
integrate a history of river, catchment, and oceanic variability (Hedges et al, 1997).
Terrestrial OM can be differentiated from marine OM using carbon to nitrogen (C/N) ratios and
the bulk carbon isotopic composition ($^{13}$C) of sedimentary OM (e.g. Meyers, 1994). The
abundance of N-free macromolecules such as lignin or cellulose result in organic carbon-rich
plant tissues that lead to an overall higher C/N ratio for terrestrial OM compared to aquatic
organisms (Hedges et al., 1986). However, this ratio may be biased when plant-tissues gain
nitrogen during bacterial degradation and when planktonic OM preferentially lose nitrogen over
carbon during decay (Hedges and Oades, 1997). Differences in the stable carbon isotopic
composition may also be used to examine terrestrial input as terrestrial OM is typically depleted
in $^{13}$C ($\delta^{13}$C of -28 to -25‰) compared to marine OM (-22 to -19‰). However, C$_4$ plants have
$\delta^{13}$C values of around -12‰ (Fry et Sherr, 1984 ; Collister et al. 1994; Rommerskirchen et al.,
2006) and thus a substantial C$_4$ plant contribution can make it difficult to estimate the
proportion of terrestrial to marine OM in certain settings (Goñi et al., 1997).
Biomarkers of terrestrial higher plants are also used to trace terrestrial OM input into marine
sediments. For example, plant leaf waxes such as long-chain *n*-alkanes are transported and
preserved in sediments (Eglinton and Eglinton 2008, and references cited therein) and can



provide information on catchment integrated vegetation or precipitation changes (e.g. Ponton
et al., 2014), while soil specific bacteriohopanepolyols (BHP) are biomarkers of soil bacteria and
indicate changes in soil transport (Cooke et al., 2008). Similarly, branched glycerol dialkyl
glycerol tetraethers (brGDGTs) are widespread and abundant in soils (Weijer et al., 2007, 2009)
and can be used to trace soil OM input into marine settings via the branched and isoprenoid
tetraether (BIT) Index (Hopmans et al., 2004). However, brGDGTs can also be produced in-situ in
rivers (De Jonge et al., 2015) and thus the BIT index does not reflect soil OM input only.
Moreover, because the BIT index is the ratio of brGDGTs to crenarchaeol (an isoprenoidal GDGT
predominantly produced by marine Thaumarchaeota; Sinninghe Damsté et al., 2002), the BIT
index can also reflect changes in marine OM productivity instead of changes in terrestrial OM
input in areas where primary productivity is highly variable, i.e. where the quantity of
crenarchaeol is variable (Smith et al., 2012).
Although these terrestrial organic proxies are useful to trace soil, river or vegetation input into
marine sediments thus far there are no organic geochemical proxies to specifically trace riverine
OM input. However, recently, the $C_{32}$ 1,15-diol was proposed as a tracer for riverine OM input
(De Bar et al., 2016; Lattaud et al., 2017). This diol, together with other 1,13 and 1,15-diols, are
likely derived from freshwater eustigmatophyte algae (Volkman et al., 1999; Rampen et al.,
2007, 2014b; Villanueva et al., 2014). Versteegh et al. (2000) showed that the proportion of $C_{32}$
1,15-diol to other diols was relatively higher closer to the mouth of the Congo River. Likewise,
Rampen et al. (2012) observed that sediments from the estuarine Hudson Bay have a much
higher proportion of $C_{32}$ 1,15-diol than open-marine sediments. More recent studies noted
elevated amounts of the $C_{32}$ 1,15-diol in coastal sediments, and even higher amounts in rivers



indicating a continental source for this diol (De Bar et al., 2016, Lattaud et al., 2017). Since the
$C_{32}$ 1,15-diol was not detected in soils distributed worldwide, production of this diol in rivers by
freshwater eustigmatophytes is the most likely source of this compound which, therefore, can
potentially be used as a proxy of riverine OM input to marine settings.
Here we test the downcore application of this new proxy by analysing the fractional abundance
of the $C_{32}$ 1,15-diol in a shelf sea record (0-45 ka) from the Mozambique Channel and a record
(0-24 ka) from the Eastern Mediterranean Sea to reconstruct Holocene/Late Pleistocene
changes in freshwater input of the Zambezi and Nile rivers, respectively. Analysis of surface
sediments and comparision with previously published BIT index records (Castañeda et al., 2010;
Kasper et al., 2015) allows us to assess the potential of the $C_{32}$ 1,15-diol as a tracer for riverine
runoff in these coastal margins.

**2. Material and Methods**
*2.1. Study sites*
*2.1.1. Mozambique margin and Zambezi River*
The Mozambique Channel is located between the coasts of Mozambique and Madagascar
between 11°S and 24°S and it plays an important role in the global oceanic circulation by
transporting warm Indian Ocean surface waters into the Atlantic Ocean. The Zambezi River is
the largest river that delivers freshwater and suspended particulate matter to the Mozambique
Channel (Walford et al., 2005). The Zambezi River has a drainage area of $1.4 \times 10^{6}$ km$^2$ and an
annual runoff between 50 and 220 km$^3$ (Fekete et al., 1999). It originates in northern Zambia,



95 flows through eastern Angola and Mozambique to reach the Indian Ocean. The Zambezi delta

96 starts at Mopeia (Ronco et al., 2006) and the Zambezi plume enters the Mozambique Channel

97 and flows northwards along the coast (Nehama and Reason, 2014). The rainy season in the

98 catchment is in austral summer when the Intertropical convergence zone (ITCZ) is at its

99 southernmost position (Beilfuss and Santos, 2001; Gimeno et al., 2010; Nicholson et al., 2009).

100 The seasonal variation of the Zambezi runoff varies between 7000 $m^3$/s during the wet season

101 to 2000 $m^3$/s during the dry season (Beilfuss and Santos, 2001). A few smaller Mozambique

102 rivers other than the Zambezi River flow into the Mozambique Channel: the Ligonha, Licungo,

103 Pungwe and Revue in Mozambique (together with the Zambezi River, they are collectively called

104 "the Mozambique rivers" here).

105 Past studies have shown that the deposition pattern of the Zambezi riverine detritus is variable

106 with sea level, i.e. most of the time material was deposited downstream of the river mouth but

107 during high sea level it was deposited northeast of the river mouth due to a shore current

108 (Schulz et al., 2011). During the last glacial period the Zambezi riverine detritus followed a more

109 chanellized path (Schulz et al., 2011). Van der Lubbe et al. (2016) found that the relative

110 influence of the Zambezi river compared to more northern rivers in the Mozambique Channel

111 varied during Heinrich event 1 (H1) and the Younger Dryas (YD). Schefuss et al. (2011) studied

112 the $\delta^{13}C$ and $\delta D$ of *n*-alkanes, and the elemental composition (Fe content) of core GeoB9307-3,

113 located close to the present day river mouth (Fig. 1), and reported higher precipitation and

114 riverine terrestrial input in the Mozambique Channel during the Younger Dryas and H1. This is in

115 agreement with more recent results from Just et al. (2014) on core GeoB9307-3 and Wang et al.



(2013a) on core GIK16160-3, further away from the actual river mouth; both studies also
showed an increased riverine terrestrial input during H1 and YD.

*2.1.2 Eastern Mediterranean Sea and Nile River*
The Eastern Mediterranean Sea is influenced by the input of the Nile River, which is the main
riverine sediment supply with annual runoff of 91 km$^3$ and a sediment load of about 60 x 10$^9$
kg.yr$^{-1}$ (Foucault and Stanley, 1989; Weldeab et al., 2002). Offshore Israel, the Saharan eolian
sediment supply is very low (Weldeab et al., 2002). A strong north-eastern current distributes
the Nile River sediment along the Israeli coast toward our study site. The Nile River consists of
two main branches: the Blue Nile (sourced at Lake Tana, Ethiopia) and the White Nile (sourced
at Lake Victoria, Tanzania, Uganda). Precipitation in the Nile catchment fluctuates widely with
latitude with the area north of 18°N dry most of the year and the wettest areas at the source of
the Blue Nile and White Nile (Camberlin, 2009). This general distribution reflects the latitudinal
movement of the ITCZ.
Castañeda et al. (2010) have shown that sea surface temperature (SST) (reconstructed with
alkenones and TEX$_{86}$) at the study site was following Northern Hemisphere climate variations
with a cooling during the Last Glacial Maximum (LGM), Heinrich event 1 (H1) and Younger Dryas
(YD) and warming during the early part of the deposition of sapropel 1 (S1). Associated with the
cooling of H1 and the LGM, extreme aridity in the Nile catchment is observed as inferred from
the δD of leaf waxes, in contrast to the time of Early Holocene S1 deposition, which corresponds
to a more humid climate and enhanced Nile River runoff (Castañeda et al., 2016). Neodymium



($\varepsilon_{Nd}$) and strontium ($^{87}Sr/^{88}Sr$) isotopes (Castañeda et al., 2016; Box et al., 2011, respectively)
show an enhanced contribution of Blue Nile inputs when the climate is arid (H1, LGM) and an
increased contribution of the White Nile when the climate is humid (S1). This change also affect
the soil input into the Nile River, as inferred from the distribution of branched GDGTs, with a
more arid climate reducing the vegetation in the Ethiopian highlands (source of the Blue Nile)
and favoring soil erosion while during a more humid climate, vegetation increasing and soil
erosion is less (Krom et al., 2002).
*2.2. Sampling and processing of the sediments*
*2.2.1. Mozambique Channel sediments*
We analyzed 36 core-top sediments (from multi cores) along a transect from the Mozambique
coast to Madagascar coast (LOCO transect, Fallet et al. 2012). The LOCO core-tops have been
previously studied by XRF and grain-size analysis (van der Lubbe et al., 2014, 2016) as well as for
inorganic ($\delta^{18}O$, Mg/Ca) and organic (TEX$_{86}$, Uk'$_{37}$) temperature proxies (Fallet et al., 2012). 25
core-top sediments (from grabs, gravity or trigger-weight corers) retrieved during the R/V
Valdivia's Expeditions VA02 (1971) and VA06 (1973) (called VA for the rest of this study, Schulz
et al., 2011), comprising a north-south transect paralleling the East African coast, and spanning
from 21°S to 15°N (Fig. 1a) were also analyzed. These surface sediments have been studied
previously for element content (TOC, TON), isotopic content ($\delta^{18}O$, $\delta^{13}C$) as well as for mineral
and fossil (foraminifera) content (Schulz et al., 2011). Piston core 64PE304-80 was obtained
from 1329 m water depth during the INATEX cruise by the RV Pelagia in 2009 from a site
(18°14.44'S, 37°52.14'E) located on the Mozambique coastal margin, approximately 200 km

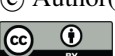



north of the Zambezi delta (Fig. 1a). The age model of core 64PE304-80 is based on $^{14}$C dating of
planktonic foraminifera (van der Lubbe, 2014; Kasper et al., 2015) and by correlation of log
(Ti/Ca) data from XRF core scanning with those of nearby core GIK16160-3, which also has an
age model based on $^{14}$C dating of a mixture of planktonic foraminifera (see van der Lubbe et al.,
2014 for details).
The LOCO sediment core-tops were sliced into 0 - 0.25 and 0.25 - 0.5 cm slices and extracted as
described by Fallet et al. (2012). Briefly, ultrasonic extraction was performed (x 4) with a solvent
mixture of dichloromethane (DCM)/methanol (MeOH) (2 : 1 v/v). The total lipid extract (TLE)
was then run through a $Na_2SiO_4$ column to remove water. The 25 VA core-tops from the
Valdivia's expedition were freeze dried on board and stored at 4 °C. They were extracted via
Accelerator Solvent Extractor (ASE) using DCM: MeOH mixture 9:1 (v/v) and a pressure of 1000
psi at 100 °C using three extraction cycles.
We analyzed sediments of core 64PE304-80 for diols using solvent extracts that were previously
obtained for determination of the BIT index and δD ratio of alkenones (Kasper et al., 2015).
Briefly, the core was sliced into 2 cm thick slices and the sediments were ASE extracted using
the method described above.
For all Mozambique Channel sediments, the total lipid extract (TLEs) were separated through an
alumina pipette column into three fractions: apolar (Hexane : DCM, 9:1 v/v), ketone (Hexane :
DCM, 1:1 v/v) and polar (DCM : MeOH, 1:1 v/v). The polar fractions, containing the diols and
GDGTs, were dissolved into a mixture of 99:1 (v/v) Hexane : Isopropanol and filtered through a
0.45 µm PTFE filters.



*2.2.2. Eastern Mediterranean sediment core*
Gravity core GeoB 7702-3 was collected during the R/V Meteor cruise M52/2 in 2002 from the
slope offshore Israel (31°91.1'N, 34°04.4'E) at 562 m water depth (Castañeda et al., 2010). The
chronology of this sedimentary record is based on 15 planktonic foraminiferal $^{14}$C AMS dates
(Castañeda et al., 2010). The sediments have previously been analyzed for GDGTs, alkenones,
δD and δ$^{13}$C of leaf wax lipids, and bulk elemental composition (Castañeda et al., 2010, 2016).
Sediments were sampled every 5 cm and 1 cm thick, and previously extracted as described by
Castañeda et al. (2010). Briefly, the freeze-dried sediment were ASE extracted and the TLEs
were separated using an aluminum oxide column into 3 fractions as described above.
*2.3. Analysis of long-chain diols*
Diols were analyzed by silylation of the polar fraction with 10 μL N,O-Bis(trimethylsilyl)-
trifluoroacetamide (BSTFA) and 10 μL pyridine, heated for 30 min at 60°C and adding 30 μL of
ethyl acetate. Diol analysis was performed using a gas chromatograph (Agilent 7990B GC)
coupled to a mass spectrometer (Agilent 5977A MSD) (GC-MS) and equipped with a capillary
silica column (25 m x 320 μm; 0.12 μm film thickness). The oven temperature regime was as
follows: held at 70 °C for 1 min, increased to 130 °C at 20 °C/min, increased to 320 °C at 4
°C/min, held at 320 °C during 25 min. Flow was held constant at 2 mL/min. The MS source
temperature was held at 250 °C and the MS quadrupole at 150 °C. The electron impact
ionization energy of the source was 70 eV. The diols were quantified using selected ion
monitoring (SIM) of ions m/z 299.4 (C$_{28}$ 1,14), 313.4 (C$_{28}$ 1,13, C$_{30}$ 1,15), 327.4 (C$_{30}$ 1,14), and
341.4 (C$_{32}$ 1,15) (Versteegh et al., 1997; Rampen et al., 2012).



The fractional abundance of the $C_{32}$ 1,15-diol is expressed as percentage of the total major diols
as follows:
$$FC_{32}1,15 = \frac{[C_{32}1,15]}{[C_{28}1,13]+[C_{28}1,14]+[C_{30}1,13]+[C_{30}1,14]+[C_{30}1,15]+[C_{32}1,15]} \times 100 \qquad (1)$$

*2.4. Analysis of GDGTs*
GDGTs in the polar fractions of the extracts of the VA and LOCO core-top sediments were
analyzed on an Agilent 1100 series LC/MSD SL following the method described by Hopmans et
al. (2016). The BIT index was calculated according to Hopmans et al. (2004). We calculated the
#ring tetra as described by Sinninghe Damsté et al. (2016) and the CBT index and soil pH as
described by Peterse et al. (2012):
$$\#ring\ tetra = \frac{GDGT\ Ib + 2 \times GDGT\ Ic}{GDGT\ Ia + GDGT\ Ib + GDGT\ Ic} \qquad (2)$$
$$CBT = \log\left(\frac{GDGT\ Ib + GDGT\ IIb}{GDGT\ Ia + GDGT\ IIa}\right) \qquad (3)$$
$$pH = 7.9 - 1.97 \times CBT \qquad (4)$$

**3. Results**
*3.1. Surface sediments of the Mozambique Channel*
$F_{1,15-C_{32}}$ in surface sediments across the Mozambique Channel varies from 2.3 to 12.5% (Fig. 1d,
1f) with one of the highest value in front of the Zambezi River mouth (10%). The core-tops
located in front of other minor northern rivers (Licungo, Ligonha Rivers) are also characterized
by values of $F_{1,15-C_{32}}$ (>7.5%) higher than those further away from the coast (< 5%). The major



diol in all Mozambique surface sediments is the $C_{30}$ 1,15-diol (57.5±9.9%) with lower amounts of
the $C_{30}$ 1,14-diol (21.1±6.0%) and $C_{28}$ 1,14-diol (13.2±4.9%) (Fig. 1f).
The values for the BIT index in surface sediments across the Mozambique Channel vary from
0.01 to 0.42 (Fig. 1c). BIT values are highest in the most northern region (0.4) and in front of
river mouths (0.2-0.3) compared to values found close to the coast of Madagascar (<0.04).
Following Sinninghe Damsté (2016), we calculated the #ring tetra (the relative abundance of
cyclopentane rings in tetramethylated branched GDGTs) to determine if the brGDGTs are in-situ
produced in the surface sediments or derived from the continent. The #ring tetra has an
average of 0.39±0.03 with higher values in front of the river mouths (with the highest values
close to the Madagascar rivers) and shows a clear decrease towards the open ocean (Fig. 1d).
The low #ring tetra indicate that there is likely limited in-situ sedimentary production of
brGDGTs in the sediments of the Mozambique coastal shelf area except for the samples closest
to the Madagascar coast where high #ring tetra values and low BIT values indicate in-situ
production of brGDGTs. However, for the Mozambique shelf, the brGDGTs are mostly derived
from the continent, confirming the use of the BIT index as a tracer for freshwater input in this
region.
*3.2. Holocene and Late Quaternary sediments of the Mozambique Channel and Nile River*
In the sediments of the Mozambique Channel core 64PE304-80, $F_{1,15\text{-}C32}$ shows a wide range; it
varies from 2.4 to 47.6% (Fig. 2). Between 44 and 39 ka the values are relatively stable (average
of 27.6 ± 4.5%), then they rapidly decline between 39 and 36 ka to 11%. From this point on they
gradually increase, reaching 37.4% at 17 ka. $F_{1,15\text{-}C32}$ is then rapidly decreasing until it reaches the



lowest values of the record after 12 ka (average of 4.9 ± 1.4%). Holocene sediments (0-11 ka)
show relatively low and constant values of $F_{1,15-C32}$ (5± 1.5%), similar to the values found in the
surface sediments of the area, i.e. 3.5 ± 1.6% (Figs. 1 and 2d).
The BIT index record shows similar changes (data from Kasper et al., 2014) as that of $F_{1,15-C32}$.
Between 44 and 39 ka the average BIT value is 0.43±0.06, then the BIT value decreases to 0.36
at 36 ka, followed by an increase until 17 ka to reach 0.6, while the Holocene values are
constant (average 0.1±0.02). The #ring tetra of branched GDGTs is constantly low (average
0.15±0.01; Fig. S1a) between 44 to 15.5 ka, then increases to 0.4 at 8 ka and stays constant until
the end of the Holocene (average 0.34±0.03). Overall, these values are low and do not approach
the values (0.8-1.0) associated with in-situ production of branched GDGTs in coastal marine
sediments (Sinninghe Damsté, 2016). The #ring tetra also shows a negative correlation with the
BIT index throughout the record ($R^2$=0.74, $p<0.05$), indicating that when BIT values are high,
#ring tetra is low. Therefore, high BIT values can definitely be associated with terrestrial brGDGT
input. If we assume the in-situ production of brGDGTs in the river (e.g. DeJonge et al., 2015; Zell
et al., 2015) is minimal, we can then infer sources of soils from the different catchment areas by
reconstructing the soil pH via the CBT index (see equation 3 and 4, Peterse et al., 2012). This
showed a constant soil pH (average 6.2±0.1) from 43 to 15 ka followed by a slight increase to 7
at 8 ka and constant (average 6.8±0.08) at the end of Holocene (Fig. S1b).
In Eastern Mediterranean sediment core GeoB 7702-3, $F_{1,15-C32}$ ranges from 3.9 to 47.0%.
Between 24 and 15 ka the values are slowly decreasing from 41% at 24 ka to 7% at 15 ka.
Subsequently, $F_{1,15-C32}$ raises sharply until 11.7 ka (44%) followed by a sharp decrease down to
16% at 10 ka. $F_{1,15-C32}$ increases again until 7.5 ka up to 30%, followed by a slow decrease in the



Late Holocene towards values as low as 6% (Fig. 3a). The BIT index (data from Castañeda et al.,
2016) varies similar to $F_{1,15\text{-}C32}$. It is constant between 24 and 17 ka (average 0.37±0.05), then
decreases to 0.13 at 14.5 ka. It subsequently increases between 15.6 and 9 ka, before
decreasing after 9 ka and stays constant in the Holocene (average 0.17±0.05). The #ring tetra of
the brGDGTs (Fig. S1c) is constant from 24 to 15 ka (0.37±0.05) then shows lower values from 15
to 7 ka (0.29±0.04) and, finally, increases again during the late Holocene (0.40±0.05). The BIT
index and #ring tetra do not show a clear negative correlation as observed for the Mozambique
core. However, the values of #ring tetra are well below 0.8-1.0, suggesting that in-situ
production of brGDGTs does not play an important role, in line with the depth from which the
core was obtained which is well below the zone of 100-300 m where in-situ production is most
pronounced (Sinninghe Damsté, 2016). During parts of the record, low #ring tetra are associated
with high BIT values, indicating that between 24 and 7 ka the brGDGT are mainly terrigenous.
For the oldest part of the core, the soil pH shows a stable period from 24 to 14.8 ka (average
6.94±0.07) then increases to 7.3 at 15 ka, followed by a large decrease (pH reaching 6.5 at 8.5
ka). As the in-situ production of brGDGT is likely to be minimal in the latest part of the
Holocene, the soil pH can be reconstructed via the CBT index and shows a stable pH (average of

279  6.8±0.1).

**4. Discussion**
*4.1. Application of C$_{32}$ 1,15-diol as a proxy for riverine input in the Mozambique shelf.*
The percentage of the C$_{32}$ 1,15-diol is overall relatively low (<10%) in the surface sediments of
the Mozambique Channel in comparison with other coastal regions with a substantial river input





(Fig. 1f), where values can be as high as 65% (De Bar et al., 2016; Lattaud et al., 2017).
Moreover,  the BIT values are also relatively low at 0.01-0.42. Further confirmation of the low
amount of terrestrial input in the analyzed surface sediments comes from the low C/N values
(between 4.2 and 8.9 for the VA surface sediments; Schulz et al., 2011), characteristic of low
terrestrial OM input (Meyers 1994). Nevertheless, the slightly higher values of both the BIT
index and the $F_{1,15-C32}$ near the river mouths indicate that both proxies do seem to trace present
day riverine input into the Mozambique  Channel, in line with earlier findings of other coastal
margins influenced by river systems (De Bar et al, 2016; Lattaud et al., 2017).

*4.2. Past variations in riverine input in the Mozambique Channel*
We compared  the record  of $F_{1,15-C32}$ with previously published proxy records, in particular the
BIT index (Kasper et al., 2015) and log (Ca/Ti) (van der Lubbe et al., 2016). These two proxies
show the same pattern as $F_{1,15-C32}$ (Fig. 2). Indeed, the BIT index and the percentage of $C_{32}$ 1,15-
diol are strongly correlated ($r^2$ =0.83, p<0.001). Since the #ring tetra of brGDGTs varies between
0.06 and 0.4 (Fig. S1a) and is significantly negatively correlated with the BIT values, the brGDGTs
are predominantly derived from the continent (cf. Sinninghe Damsté, 2016) and thus the BIT is
likely reflecting terrigenous input in the marine environment. Furthermore,  the percentage of
$C_{32}$ 1,15-diol also shows a significant negative correlation with log(Ca/Ti) ($r^2$=0.43, p<0.0001,
van der Lubbe et al., 2016). This is another proxy for riverine input since Ti is mainly derived
from erosion of continental rocks transported to the ocean through rivers, whereas Ca derives
predominantly  from the marine environment.





The records of $F_{1,15-C32}$ and BIT index show three major variations: a steep drop from 19 to 10 ka,
a slow increase from 38 to 21 ka during the Last Glacial Stage and a steep decrease between 40
to 38 ka. The largest change in the BIT index and $F_{1,15-C32}$ is between 19 to 10 ka, i.e. a major
drop which coincides with an interval of rapid sea level rise (Fig. 2b). Following Menot et al.
(2006), we explain the drop in the BIT index, and consequently also the drop in $F_{1,15-C32}$, by the
significant sea level rise occurring during this period. Rising sea level flooded the Mozambique
plateau, moving the river mouth further away from the core site and establishing more open-
marine conditions. This most likely resulted in lower $F_{1,15-C32}$ and BIT values, conditions that
remained throughout the Holocene. The decrease in the delivery of terrestrial matter is also
seen in element ratios (Fe/Ca) and organic proxies (BIT) in nearby core GeoB9307-3 (Schefuß et
al, 2011), which is located closer to the present day river mouth in the Mozambique plateau
(Fig. 1a). Likewise, the gradual increase in the BIT index and $F_{1,15-C32}$ between 38 and 21 ka
occurred at a time when sea-level was decreasing (Fig 2b., Grant et al., 2014; Rohling et al.,
2014) and thus the river mouth came closer to our study site. Furthermore, between 38 and 35
ka there is also an increase in precipitation in the catchment as reconstructed by the δD of *n*-
alkanes in surrounding sediment cores (Tierney et al., 2008; Schefuß et al., 2011; Wang et al.,
2013a; Fig. 2d). A wetter period may be characterized by increased erosion and a higher river
flow, which could bring more $C_{32}$ 1,15-diols and brGDGTs into the marine realm. The decrease of
BIT values and $F_{1,15-C32}$ during 40-38 ka coincides with Heinrich event 4 (H4), a cold and dry event
in this part of Africa (Partridge et al., 1997; Tierney et al., 2008; Thomas et al., 2009), with dry
conditions perhaps leading to a reduced riverine input into the ocean and also a reduced input
of brGDGTs and the $C_{32}$ 1,15-diol.





Interestingly, there are two periods where BIT and $F_{1,15-C32}$ records diverge (Fig. 2a): during the
Younger Dryas (YD; 12.7-11.6 ka) and Heinrich event 1 (H1; 17-14.6 ka) with the BIT index
decreasing ca. 1 ky later than $F_{1,15-C32}$. Comparison with the Ca/Ti ratio shows that both during
H1 and the YD, the Ca/Ti ratio increased at the same time as the $C_{32}$ 1,15-diol but earlier than
the BIT index, suggesting that the latter was influenced by other parameters. The BIT index is
the ratio of brGDGTs (produced mostly in soil or in-situ in rivers in our area based on the low
values for #ring tetra; Sinninghe Damsté, 2016) over crenarchaeol (produced in marine or
lacustrine environments; Schouten et al., 2013 and references cited therein). As both the Ti/Ca
ratio and $F_{1,15-C32}$ indicate a decrease in riverine input, a constant BIT index can be explained by
two options: a simultaneous decrease in crenarchaeol (marine) production or a change in soil
input with higher brGDGT concentrations eroding into the river. The concentration of
crenarchaeol during H1 is relatively stable but there is a slight decrease of crenarchaeol during
YD (Fig. S2b). Thus, the difference between BIT and $F_{1,15-C32}$ during YD can be partly explained by
decreased crenarchaeol production together with a decrease in branched GDGTs due to a
reduced river input leading to relatively stable BIT values. In contrast, crenarchaeol and brGDGT
concentrations are relatively stable during H1 and thus the lower river influx, as indicated by the
Ca/Ti and $F_{1,15-C32}$, apparently did not lead to a decrease in brGDGT input. This could be due to a
shift of sources of soil which are eroded in the river, i.e. if in this period there is a shift towards
soils with relatively higher brGDGT concentrations, the BIT index would remain high despite a
decrease in riverine input.
A shift in soil sources may be due to two major changes that happened during this period (and
also during the YD), i.e. a shift in catchment area of the Zambezi River (Schefuß et al., 2011, Just





et al., 2014) and a shift in the relative influence of the Zambezi River versus northern
Mozambique rivers (van der Lubbe et al., 2016). The shift in catchment area is evident from the
higher influx of kaolinite-poor soil into the marine system during H1 and YD (Just et al., 2014)
coming from the Cover Sands of the coastal Mozambique area (Fig. 3d, blue circle), relative to
the kaolinite-rich soils of the hinterlands (Fig. 3d, red circles). If the brGDGT concentrations from
the latter region are higher, then this change of soil input could lead to a stable brGDGT flux into
the marine environment, despite decreasing Zambezi River runoff. Support for a shift in soil
sources comes from the soil pH record reconstructed from brGDGTs, which during the YD shows
a shift towards more acidic soils. However, no changes in soil pH are observed during H1.
The relative influence of other rivers (Lurio, Rovuma Rivers) relative to the Zambezi River (Fig.
3d green circle) was inferred from neodymium isotopes by Van der Lubbe et al. (2016), i.e. more
radiogenic rocks are found in the northern river catchments in comparison to the rocks in the
Zambezi catchment (Fig. 2b). These authors found that during H1 and YD, the relative
contribution of the northern rivers is lower than normal, likely due to drought conditions north
of the Zambezi catchment area (Tierney et al., 2008, 2011; Just et al., 2014). These northern
rivers run through a catchment containing mainly humid highstand soils, which are different soil
types than observed in the catchment area of the Zambezi River (van der Lubbe et al., 2016).
Higher brGDGT concentrations in the soils of the catchment areas of the Zambezi River can
potentially explain the discrepancy between BIT and $F_{1,15\text{-}C32}$, i.e. during H1 and YD there is more
input of brGDGT-rich soils from the Zambezi than brGDGT-poor soils from the northern rivers
leading to constant BIT values despite a dropping riverine input. Further research examining the



brGDGT contents of soils in the different river catchment areas is required to distinguish
between the different hypotheses.
*4.3. Past variations in riverine input in the Eastern Mediterranean Sea*
Like with the Mozambique Channel core, we compared $F_{1,15-C32}$ in core GeoB7702 with other
terrigenous proxies: BIT index, log (Ca/Ti) and strontium isotopes, the latter to infer the relative
importance of the Blue Nile and the White Nile as source regions (Fig. 4c-e). The BIT values (data
from Castañeda et al., 2010) shows a significant positive correlation with $F_{1,15-C32}$ ($r^2$=0.38, p <
0.05), while log (Ca/Ti) shows an negative correlation to $F_{1,15-C32}$, again in agreement with a
terrigenous origin of the $C_{32}$ 1,15-diol. $F_{1,15-C32}$ and BIT records show much lower Holocene
values compared to pre-Holocene (12±6% and 0.18±0.06 for the Holocene and 27±11% and
0.38±0.11 before the Holocene, respectively), which again can be linked to the sea level rise
occurring during the last deglaciation, i.e. our study site was further away from the river mouth
and the amount of terrigenous OM reaching the site decreased. Both records show low values
during H1 comparable to the Holocene. These low values can be attributed to enhanced aridity
in the Nile River catchment (Castañeda et al., 2016) leading to lower river flow and decreasing
the amount of terrigenous OM reaching our core site.
In this core, there are 3 major discrepancies observed between the BIT index and $C_{32}$ 1,15-diol:
(1) during the LGM, between 22-19 ka, where the $C_{32}$ 1,15-diol shows a decrease while the BIT
index remains constant, (2) during the onset of the deposition of S1 (6.1-10.5 ka, Grant et al.,
2016) where the BIT index decreases later than the $C_{32}$ 1,15-diol, and (3) after 2 ka when the BIT
index increases while the $C_{32}$ 1,15-diol decreases. For the LGM the percentage of $C_{32}$ 1,15-diol is



decreasing, log (Ca/Ti) is as well, but the BIT index remains constant indicating that there is no
significant decrease in terrigenous OM reaching the core site at that time. There is no significant
change in continental climate, based on the findings of Castañeda et al. (2016), suggesting no
change in vegetation cover or river flux. This suggests that the change in $F_{1,15-C32}$ is not due to a
change in the input of $C_{32}$ 1,15-diol but in other, mainly marine derived, diols, in particular the
$C_{30}$ 1,15-diol. An increase in this marine diol will lower the $F_{1,15-C32}$ but if the amount of
crenarchaeol is not changing at the same time, the BIT values will remain unaffected.
The deposition of S1 is described as a period of increased riverine input leading to stratification
and anoxia (Rossignol-Strick et al., 1982). However, an increased river input is neither reflected
in the $C_{32}$ 1,15-diol nor in the BIT index, in fact both of them are asynchronously decreasing.
Castañeda et al. (2010) showed that the decrease in the BIT index is due to a large increase in
crenarchaeol (Fig. S3b), much larger than the increase in brGDGTs, due to increased productivity
and preservation. A similar scenario may apply for the diols, i.e. the marine diols (in particular
the $C_{30}$ 1,15-diol, data not shown) are also increasing at that time more substantially than the
$C_{32}$ 1,15-diol, thus lowering the percentage of $C_{32}$ 1,15-diol. However, there is a difference in
timing, i.e. the BIT index decreases slightly later than the $C_{32}$ 1,15-diol (9.1 and 10.5 ka,
respectively). The decrease in the $C_{32}$ 1,15-diol coincides with a substantial increase in sea level
(Fig. 4b). This increase in sea level will increase the distance between the core site and the river
mouth decreasing the amount of terrigenous material reaching the site. This decrease is also
visible to some extent in the log (Ca/Ti) but not in the BIT index. Possibly, like with the
Mozambique Channel, the brGDGT fluxes in the river was much higher at that time. Indeed, the
Sr isotopes suggest a major shift from a Blue Nile to a White Nile source at this time, with the





latter possibly containing more eroded soils with high brGDGT concentrations. This shift in soil
sources is also shown in the change towards more acidic soil pH during that period based on the
CBT index (Fig. S1d).
For the most recent part of the record (0-5 ka), the BIT index increases, while the percentage of
$C_{32}$ 1,15-diol is slightly decreasing. Since log (Ca/Ti) (Fig. 4c) is decreasing at this time, it suggests
that river run off was decreasing leading to lower $C_{32}$ 1,15-diol input but apparently not to a
change in the BIT index. The $\delta D_{leaf\,waxes}$ (Fig 4.d) shows it was period of increased aridity which
was probably the cause of the decreased runoff. The reason the BIT index is increasing rather
than decreasing is due to an increase in brGDGT concentration (Fig. 3b), despite evidence for a
decrease in river runoff. This can possibly be linked to the amount of vegetation in the Nile
catchment, i.e. at that time there was a decrease in vegetation cover (Blanchet et al., 2014,
Castañeda et al., 2016) which led to more soil erosion and thus potentially a higher brGDGT flux
and a higher BIT index.
The results for the Nile core as well as those from the Mozambique Channel illustrate that the
$C_{32}$ 1,15-diol seems a suitable proxy for reconstructing past riverine input into coastal seas.
However, our interpretation of the $C_{32}$ 1,15-diol record relies on the assumption that production
of this diol in rivers is not changing with different hydroclimate fluctuations on land, something
which needs to be tested. De Bar et al. (2016) showed that the percentage of $C_{32}$ 1,15-diol in the
Tagus river in Portugal did not significantly change over the course of a year, suggesting that this
assumption might be valid.
**6. Conclusion**



We studied core-tops in the Mozambique Channel and two sediment cores, in the Mozambique
Channel, off the Zambezi River mouth and in the Eastern Mediterranean Sea, offshore the Nile
delta, to test the percentage of $C_{32}$ 1,15 diol as a proxy for riverine input into the marine realm.
The surface sediments show that the $C_{32}$ 1,15-diol traces present day riverine input into the
Mozambique Channel, supported by the BIT index. In both sediment records, the $C_{32}$ 1,15-diol is
significantly correlated with the BIT index showing the applicability of this proxy to trace riverine
input, but also showed some discrepancies. This can be explained by the different terrestrial
sources of these proxies, i.e. the BIT index is reflecting soil and riverine OM input and the $C_{32}$
1,15-diol is mainly reflecting riverine OM input. Our multiproxy approach shows that the timing
of changes in the different terrestrial proxies records can differ due to changes in catchment
area or to shifting importance of the different source rivers.
**Author contribution**
S. S. and J. L. designed the study. J. Lattaud analyzed the surface sediments for diols and GDGT and core
GeoB 7702-3 for diols, I. C. sampled and extracted the surface sediments and the sediment cores
64PE304-80 and GeoB 7702-3, D. D. analyzed the sediment core 64PE304-80 for diols. H. S. collected the
VA core-tops. J. L., S. S., I. C. and J.S. S. interpreted the data. J. L. wrote the manuscript with input of all
authors.
The authors declare that they have no conflict of interest.
**Acknowledgement**
We thanks Anchelique Mets and Jort Ossebaar for analytical help. This research has been
funded by the European Research Council (ERC) under the European Union's Seventh




Framework Program (FP7/2007-2013) ERC grant agreement [339206] to S.S., J.S.S.D. and S.S.
received financial support from the Netherlands Earth System Science Centre and this work was
in part carried out under the program of the Netherlands Earth System Science Centre (NESSC),
financially supported by the Ministry of Education, Culture and Science (OCW).

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





**Figure Legend**

**Figure 1.** Map presenting (a) the location of the core-tops (LOCO transect in orange, VA core-tops in blue) and cores (stars), (b) the mean annual salinity, from NOAA 1x1° grid (http://iridl.ldeo.columbia.edu), (c) the BIT index (LOCO transect values, VA core-tops from this study), (d) the percentage of $C_{32}$ 1,15-diol in the core-tops, (e) #ring tetra of the surface sediments (#ring tetra as defined by Sinninghe Damsté 2016), (f) Ternary diagram of $C_{28}$ (sum of $C_{28}$ 1,13 and $C_{28}$ 1,14), $C_{30}$ (sum of $C_{30}$ 1,13, $C_{30}$ 1,14 and $C_{30}$ 1,15) and $C_{32}$ ($C_{32}$ 1,15) diols (LOCO transect in orange, VA core-tops in blue, data from Lattaud et al. 2017 in purple). The maps have been draw using Ocean Data View.

**Figure 2.** Organic and lithologic proxy records for core 64PE304-80 and parallel core GIK16160-3. (a) BIT index indicating soil and riverine input (Kasper et al., 2015) and percentage of $C_{32}$ 1,15-diol tracing riverine input (b) Red Sea Level changes (Grant et al., 2013) (c) log(Ca/Ti) indicating terrestrial input (van der Lubbe et al., 2013), (d) reconstruction of δD precipitation based on leaf wax n-$C_{29}$ alkane of core GIK16160-3 (Wang et al., 2013), € $ε_{Nd}$ signatures of the clay fraction document changes in riverine influence (van der Lubbe et al., 2016). The grey bars show the Younger Dryas (YD) and Heinrich event 1 (H1) and 4 (H4).

**Figure 3.** Sources of riverine input in both area, (a) Location of core GeoB7702-3 (b) Close up location of core GeoB7702-3 and core 9509 (Box et al., 2011) (c) source of the Nile river sediments, Blue Nile: BN, White Nile: WN, Lake Tana: LT, Lake Victoria: LV (from Castañeda et al., 2016) and (d) the Mozambique Channel (red circles shows source areas of the Zambezi river during dry conditions, blue circle shows source area of the Zambezi river during wet conditions



(Just et al., 2014), and green circle show northern rivers source area (van der Lubbe et al.,

677   2016)).

**Figure 4.** Organic and lithologic proxy records for core GeoB7702-3 and core 9509. (a) BIT index
indicating soil and riverine input (Castañeda et al., 2010) and percentage of $C_{32}$ 1,15-diol tracing
riverine input (b) Red Sea Level changes (Grant et al., 2013) (c) log(Ca/Ti) indicating terrestrial
input (Castañeda et al., 2016), (d) reconstruction of δD precipitation based on leaf wax n-$C_{31}$
alkane (Castañeda et al., 2016), (e) $^{87}$Sr/$^{88}$Sr signatures of the sediment core 9509 (offshore the
Israeli coast) document changes in riverine influence (Box et al., 2011). The grey bars show the
sapropel layer (S1), Younger Dryas (YD), Heinrich event 1 (H1) and the Last Glacial Maximum
(LGM).





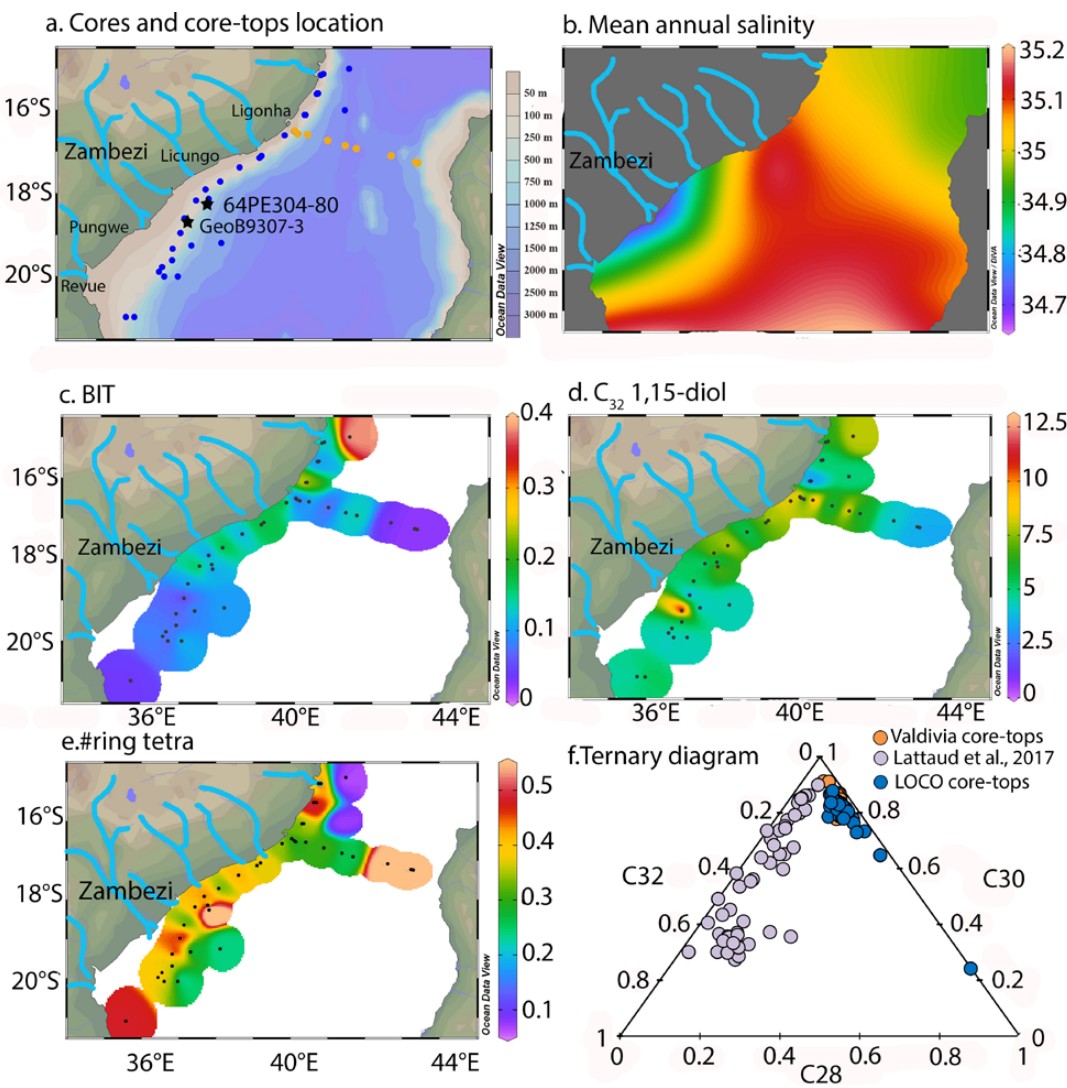


Figure 1



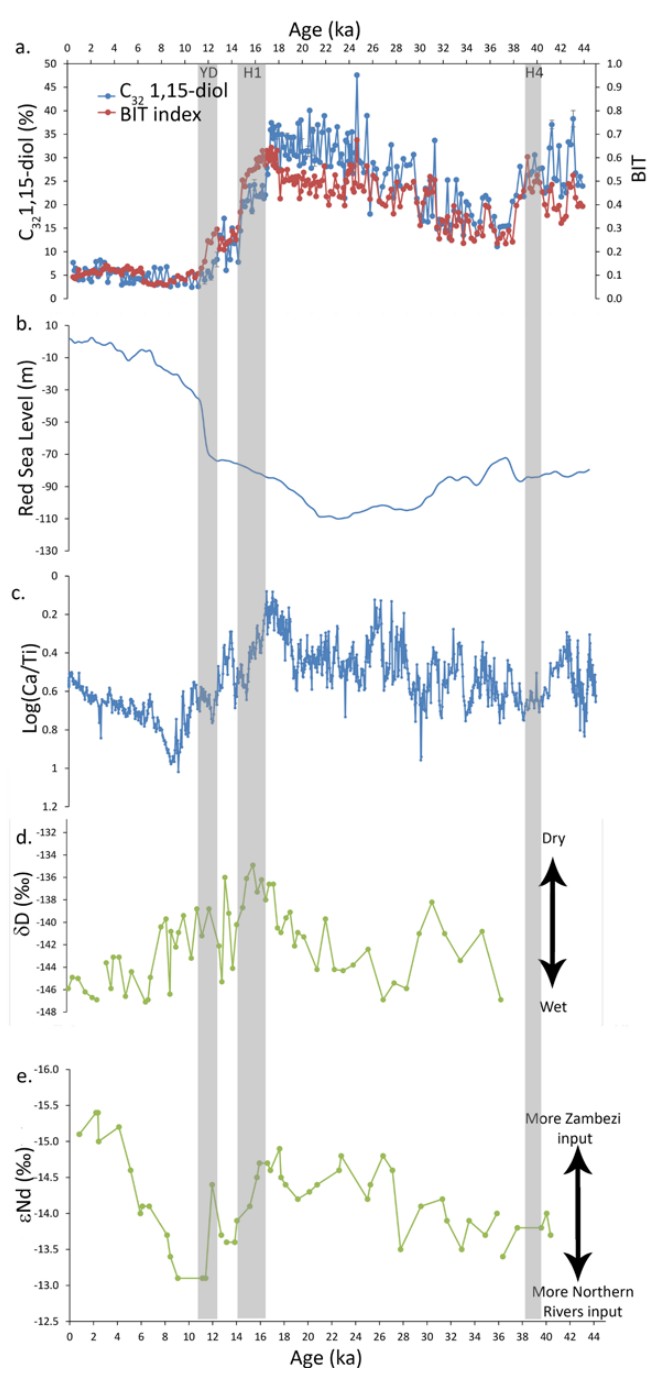

Figure 2



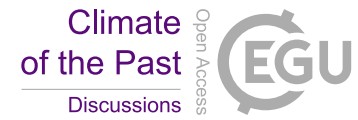

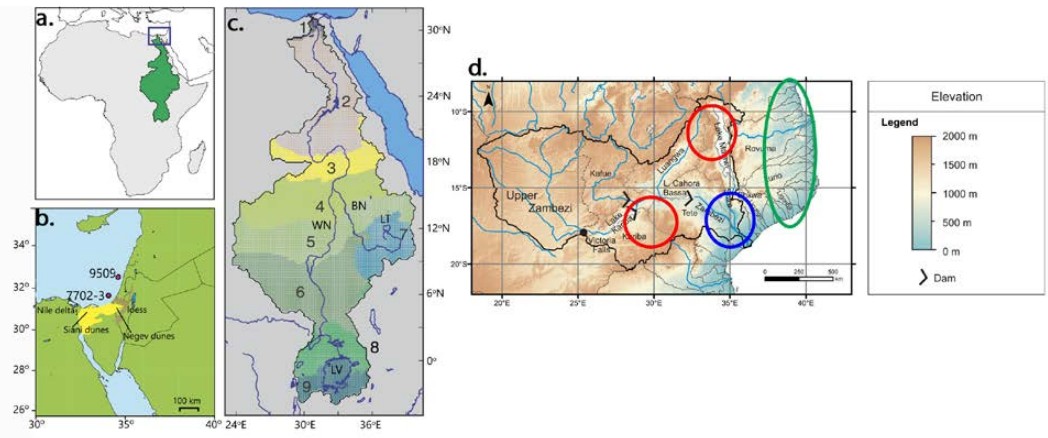



716        Figure 3


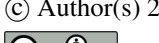



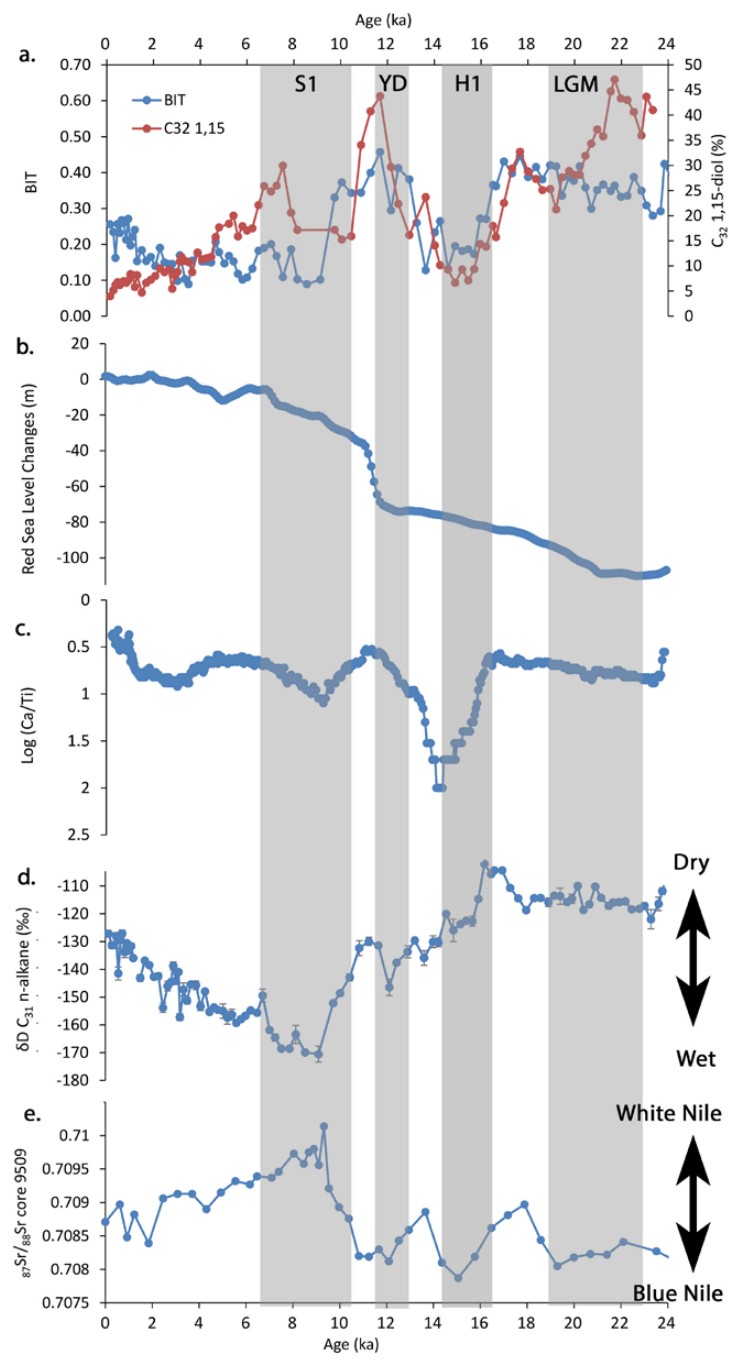


Figure 4