# Peer review of "The C32 alkane-1,15-diol as a proxy of late Quaternary riverine input in coastal margins"

_Climate of the Past, 2017_

## Short Comment (SC1) · 7 Apr 2017

Review of Lattaud et al. 2017

Accept with Major Revisions

Summary of the paper:

Lauttaud et al. tested if the fractional abundance of the C32 alkane 1,15-diol, compared to all other 1,13- and 1,15-diols, could be used as a proxy for riverine OM input in paleo work. Their proxy was ground truthed by comparing it to the BIT index of core top samples collected from the Mozambique Channel. Once they found a good correlation between the two proxies, the authors then applied their diol proxy to Quaternary sediments found off the Zambezi and Nile Rivers. Differences between the new diol

proxy and the BIT index were attributed to changes in either the sourcing of the soil OM to the river and/or changes in crenarchaeol abundance. Overall, the authors suggest that the C32 alkane 1,15-diol proxy is an adequate tracer for riverine OM inputs and that it is not as affected by soil and vegetation changes in the river's catchment area.

The main criticisms for this paper are as follows:

- The authors compare their diol proxy to the BIT index in order to determine if it traces riverine OM inputs; however, other proxies that better trace terrestrial OM inputs (e.g. lignin or long chain fatty acids) would be better to use. As stated in line 61 and throughout their discussion, the BIT index can also reflect changes in marine OM productivity.

- The discussion is focused more on how well the diol proxy compared to other proxies as opposed to how the riverine OM inputs may have responded to changes in climate. Some of the material they discuss in the methods and materials section should have been also included in their discussion, including the ITCZ's influence on both sites.

- The authors also do not address the differences in hydrology of the river systems they are analyzing. The reason for why the BIT index and the new diol index are so different for the Nile may be due to the longer length of the Nile River compared to the Zambezi River.

- Line 319: Authors mention there is an increase in precipitation in the catchment area between 38 and 35 kyr but the dD (Fig. 2D) appears to become more enriched during this time, which would indicate there are drier conditions.

- Line 383: The authors attributed the decrease in diol and brGDGT inputs during H1 for the Nile to drier conditions with reduced river flow. They then use the arid explanation again for the Nile River inputs from 0-5kyr; however, they mentioned the increase in soil input was related to less vegetation and less soil stabilization (line 422). Data/information should be provided for vegetation changes during H1 in the Nile or else the two different explanations will contradict each other.

- Line 417: Authors mention that the log(Ca/Ti) in figure 4 is decreasing over the 0-5ka record and that it indicates that there is less riverine OM input. This should actually be the reverse. If the value is lower, it would mean there is more riverine input, which throws off their interpretation for this time period.

- Line 421: Fig. 3b should be Fig. S3b

- Figures 2 and 4: Why flip the log(Ca/Ti) axis?

- Figure 4: Looks like the BIT index follows the log(Ca/Ti), which the authors describe as a proxy for riverine input, more than the proposed diol index

- Do the authors have data for the abundances of the marine 1,13- and 1,15- diols? They barely discuss how changes in the production of marine diols influence their proxy.

END of REVIEW

---

## Referee Comment (RC1) · Anonymous Referee #1 · 10 Apr 2017

The comment was uploaded in the form of a supplement:
http://www.clim-past-discuss.net/cp-2017-43/cp-2017-43-RC1-supplement.pdf

---

## Referee Comment (RC2) · Anonymous Referee #2 · 1 May 2017

Using well-studied cores and surface sediments retrieved from the Mozambique Channel and the eastern Mediterranean, the authors tested the abundance of C32 alkane-1,15,-diol as a proxy of riverine organic matter by comparing with relatively established proxies indicative of terrestrial contribution. The authors assume that the fractional abundance of C32 alkane-1,15,-diol in total C28, C30, and C32 diols represents the contribution of the organic matter produced in river and lake freshwater, in contrast to other terrestrial proxies based on land plant organic matter and soils. This is a quite unique proxy and must be tested in the application to paleoenvironmental studies.

This study benefits from previous studies conducted by the authors' and other groups. The results of the previous studies are not described well in this paper, probably to avoid duplications. I, however, think that the following information is helpful for readers to understand this paper. 1) Brief history of C28, C30, and C32 diols and their

potential source. Expand the paragraph 64-77. 2) Age controls in both Mozambique and Mediterranean cores. Indicate the control points in Figs 2 and 4 and the short description of age-depth model in supplement.

I have the following comments: 1) The results of F1,15-C32 in two different cores are discussed independently, but more synthetic discussion is necessary on its advantages and disadvantages, the reason of the discrepancy between F1,15-C32 and BIT, the reason why the discrepancy in an eastern Mediterranean core was larger than that in a Mozambique Chanel core, why F1,15-C32 works better than BIT (is crenarchaeol production more affected by sea-level controlled marine production? etc). 2) Clearer discussion is necessary on the source of C32 alkane-1,15,-diol as a proxy of riverine organic matter. What does synchronous or asynchronous variation of F1,15-C32 (freshwater OM) and BIT (soil OM) means in more general sense? 3) Lines 366-371. This part is much more speculative than other parts. If this is true, low brGDGT concentration is somehow reflected in brGDGT concentration in surface sediments at the offshore of northern rivers. 4) If possible, I want to see more general discussion on the paleoclimate (precipitation in eastern Africa and ITCZ migration, etc) during H1 and YD periods.

---

## Author Comment (AC1) · 1 May 2017

Reply to the interactive comment of Dr. T. Bianchi on "The $C_{32}$ alkane-1,15-diol as a proxy of late Quaternary riverine input in coastal margins"

We thank Dr. Bianchi for his helpful comments on our manuscript. Below follows our reply to the main comments.

*-The reviewer states that we only compare the %$C_{32}$ 1,15-diol to the BIT index and that there are better proxies for riverine input that we could use, especially because the BIT index can be influenced by productivity.* The %$C_{32}$ 1,15-diol was mainly compared to the BIT index and log(Ca/Ti), as these proxies represent riverine terrigenous input. Unfortunately, it could not be compared to lignin concentrations because this was not determined in our samples and we have little original sediment left but it would be the next goal of the study of the $C_{32}$ 1,15-diol. We did not compare it to n-alkanes or long-chain fatty acids because both can also come from eolian input and not only from riverine material. This is especially true for the Nile site where it was shown that the n-alkanes can come from eolian input from the African peninsula (Blanchet et al., 2014). It is true that the BIT index is also influenced by archaeal productivity and that is why we also report the concentration of brGDGT to constrain the influence of the concentration of crenarchaeol (representing archaeal productivity) on the BIT index.

*-The reviewer says that the discussion is focused on the comparison of proxies and not on the relationship between the riverine input and the climate, especially with the change in ITCZ location.* It is true that the discussion is more focused on comparing the new proxy to others, as this was the main goal of our paper. Both regions have been intensively studied for past climate change including the ITCZ (Blanchet et al., 2014, Castaneda et al., 2009, 2010, 2011, 2016, Schefuss et al, 2011, Just et al., 2014, Tierney et al., 2008, Wang et al., 2013, Thomas et al., 2009, Box et al., 2011), including studies which have used the same sediment cores as used here. We do not want to repeat their conclusions in our manuscript. Rather, we explicitly chose these cores and regions as much is already known about the paleoclimate, which makes it easier to understand the behavior of a new proxy.

*-The reviewer indicates that part of the difference in correlation between the $C_{32}$ 1,15-diol and BIT between the sites can be due to the different hydrological setting of the rivers.* We find it difficult to explain the difference in correlation between the $C_{32}$ 1,15-diol and the BIT index by different hydrological settings and/or the length of the river alone. Rather we feel that the different correlations is due to the factors discussed in the manuscript.

*-The reviewer mentions that the δD decreases between 35 and 38ky (line 319) and does not increase leading to drier conditions and not more humid as stated in our manuscript.* The reviewer is correct, and we will delete this part of the discussion as the change in sea level is enough to explain the increase input of $C_{32}$ 1,15-diol and brGDGTs in our core.

*-The reviewer indicates that we use the same environmental factor, i.e. aridity, to explain two different observations, i.e. a decrease in riverine input and an increase in soil erosion (line 383).* Data supporting the change in vegetation and aridity during 0-5ky have been reported by Blanchet et al. (2014) and this is mentioned in the manuscript at line 419, while data supporting the extreme aridity during H1, more than during 0-5 kyr, have been reported by Castaneda et al., (2016). It might be that the extreme aridity during H1 (when both Lake Tana and Lake Victoria, the sources of the Blue and White Nile, were desiccated) led to a lack of vegetation and increased soil erosion but also substantially reduced river flow, such that net export of soil OM

was reduced. In contrast for the period between 0-5 ky, aridity is not as severe and thus the increased soil erosion combined with a moderately reduced river flow still leads to export of terrigenous OM as also shown by the relatively higher Ca/Ti (Castaneda et al., 2016).

-*The reviewer states that a decrease in log(Ca/Ti) during 0-5 ka indicates more riverine input*. It is true that log(Ca/Ti) is decreasing during 0-5ka and that it indicates more soil input, but it does not indicate per se more riverine input itself. There could be enhanced soil erosion which leads to a larger input of soil OM yet at a similar or even reduced river flow. This will be more precisely phrased.

- *The reviewer noticed that at line 421 the figure name should be S3b and not Fig. 3.* The figure name will be changed.

-*The reviewer is asking why the axis for log(Ca/Ti) has been inverted.* By flipping the log(Ca/Ti) we highlight the similarity between this proxy and the BIT and $C_{32}$ 1,15-diol, making the correspondence between the three proxies clearer.

- *The reviewer says that in fig. 4 the BIT index and log(Ca/Ti) are more similar than the %$C_{32}$ 1,15 and log (Ca/Ti).* The BIT index follows the log(Ca/Ti) better than the diol index as they both are influenced by soil erosion and the diol index is not.

-*The reviewer is asking is we have the concentration of the 1,13 and 1,14 diols.* Unfortunately, it was not possible to quantify the concentrations of the 1,13 and 1,15 diols as they were measured from long term stored archived extracts.

---

## Author Comment (AC2) · 10 May 2017

Reply to the interactive comment of anonymous reviewer #2 on "The $C_{32}$ alkane-1,15-diol as a proxy of late Quaternary riverine input in coastal margins"

We thank the reviewer for his/her helpful comments on our manuscript. Below follows our reply to the main comments.

*-The reviewer wish to have more information on LCDs (potential sources, history).* We will expand paragraph 64-77 in the introduction with more details about the LCD discovery and potential producers in marine and freshwater environments.

*-The reviewer would like to have the age control points indicated in figure 2 and 4 and a supplement with a summary of the age model.* The control points will be added to the figures (black triangle) and a brief supplementary method describing the previously published age-models of both cores will be added.

*-The reviewer wish to have a more synthetic discussion on the proxy, with more details about its advantages and disadvantages, as well as why, in the Mozambique core, the correlation between the $C_{32}$ 1,15-diol and BIT index is better than in the Nile core but also why the C32 1,15-diol works better than the BIT.* The $C_{32}$ 1,15-diol is not working 'better' than the BIT index to trace riverine input, rather in our view it simply reflects a different pool of organic carbon being transported by rivers, i.e. river-born carbon versus soil and river born carbon in case of branched GDGTs. Like the BIT index, the $F_{1,15-C32}$ may also be affected by marine productivity as we discussed at lines 336-339, 394-397 and 401-403.

*-The reviewer asks for a clearer discussion on the source of the $C_{32}$ 1,15-diol and what does the synchronicity/asynchronicity of the variation between BIT index and $C_{32}$ 1,15-diol means in a broader sense.* In our view, this question has been discussed already at lines 327-346 and 386-397 and we want to point out that for most of the records, BIT index and $C_{32}$ 1,15-diol actually agree quite well.

*-The reviewer indicates that the hypothesis at lines 366-371 is more speculative than other part and that, if true, the low brGDGT concentration of the Northern rivers would be reflected in surface sediment at the offshore of these rivers.* We agree with the reviewer that this is speculative and this is why, at lines 428-430, we recommend for future studies to analyze surface sediments offshore the Northern rivers to confirm this hypothesis.

*- The reviewer would like, if possible, to have more general discussion on the paleoclimate during H1 and YD.* We will give some more detail in discussion section on the climate for these periods, as noted in the rebuttal of reviewer 1.

---

## Author Response (AR1)

Dear Dr. McClymont,

Please find resubmitted the manuscript entitled "The $C_{32}$ alkane-1,15-diol as a proxy of late Quaternary riverine input in coastal margins" (cp-2017-43). We thank you, and the reviewers, for your positive comments and we have used these to revise and improve our manuscript. The reviewers and you requested major changes and we took it into account. Below we have listed our responses (in bold) to the reviewers concerns and suggested changes. In addition, we added a co-author, E. Schefuß, to the manuscript as he has been instrumental in obtaining core GeoB 7702-3 and was accidentally omitted from the original manuscript. He has read and commented on the revised manuscript.

We hope that with this revision we have addressed all issues and that the revised manuscript is suitable for publication in CP.

On behalf of the co-authors,

Sincerely,

Julie Lattaud

**Editor comments:**

*- The editor agrees with the reviewers that some more information is needed in explaining the sources of the diols, both the "riverine" and "marine". This relates to the point raised on what the "riverine input" really means (freshwater input? Organic matter production within the river?).*

**More details on the sources of the long chain diols have been added in the introduction (lines 69-77): "Long-chain diols (LCD), such as the $C_{32}$ 1,15-diols, are molecules composed of a long alkyl chain ranging from 26 to 34 carbon atoms, an alcohol group at position $C_1$ and mid-chain position, mainly at positions 13, 14 and 15. They occur ubiquitously in marine environments (de Leeuw et al., 1979; Versteegh et al., 1997, 2000; Gogou and Stephanou, 2004; Rampen et al., 2012, 2014; Romero-Viana et al., 2012; Plancq et al., 2015; Zhang et al., 2011 and references therein), where the major diols are generally the $C_{30}$ 1,15-, $C_{28}$ and $C_{30}$ 1,13-diols and the $C_{28}$ and $C_{30}$ 1,14-diols. In marine environments the 1,14-diols are produced mainly by Proboscia diatoms (Sinninghe Damsté et al., 2003, Rampen et al., 2007) and the 1,13 and 1,15-diol are thought to be produced by eustigmatophyte algae". Riverine input here relates mostly to the organic carbon carried by rivers and which comprises river-born and terrrestrial (soil+vegetation) carbon. The different proxies reflect different types of carbon which are carried by rivers. This has now been more precisely defined at lines 5, 7, 93.**

*-You note in your reply to reviewer 2 that in general there is good correlation between BIT and the diol index, but is that to be expected given different sources, or are you saying that high river flow for the most part brings more terrestrial organic matter? Addressing this point does not require extensive text, but just some clarity on what is being recorded by the diols.*

**Some details have been added to make the discussion clearer, especially with regards to the riverine input concept.**

*-The editor agrees with reviewer 2 also about some areas of speculation. It isn't always clear in the text when you are making inferences based on known soil properties (e.g. across regions) or if these are speculations. I recommend that in the revised version you try to use phrases such as "we hypothesize..." to flag up the areas which need more research for confirmation.*

**More clarity has been added when we are speculating with certain interpretations.**

*- Figure 3 is quite small and not easy to read in its current form. I recommend increasing its size so that the details are visible.*

**The size of Figure 3 has been increased and clarified.**

**Reviewer #1**

We thank Dr. Bianchi for his helpful comments on our manuscript. Below follows our reply to the main comments and, where applicable, how we changed the manuscript.

*-The reviewer states that we only compare the %C$_{32}$ 1,15-diol to the BIT index and that there are better proxies for riverine input that we could use, especially because the BIT index can be influenced by productivity.*

**F$_{C32\ 1,15}$ was mainly compared to the BIT index and log(Ca/Ti), as these proxies represent riverine terrigenous input. Unfortunately, it could not be compared to lignin concentrations because this was not determined in our samples and we have little original sediment left but we agree it would be a good thing to test in a future study of the C$_{32}$ 1,15-diol. We did not compare it to n-alkanes or long-chain fatty acids because both can also come from eolian input and not only from riverine material. This is especially true for the Nile site where it was shown that the n-alkanes mainly come from eolian input from the African peninsula (Blanchet et al., 2014). It is true that the BIT index is also influenced by archaeal productivity and that is why we also report the concentration of brGDGT to constrain the influence of the concentration of crenarchaeol (representing archaeal productivity) on the BIT index.**

*-The reviewer says that the discussion is focused on the comparison of proxies and not on the relationship between the riverine input and the climate, especially with the change in ITCZ location.*

**It is true that the discussion is more focused on comparing the new proxy to others, as this was the main goal of our paper. Both regions have been intensively studied for past climate change including the ITCZ (Blanchet et al., 2014, Castaneda et al., 2009, 2010, 2011, 2016, Schefuss et al, 2011, Just et al., 2014, Tierney et al., 2008, Wang et al., 2013, Thomas et al., 2009, Box et al., 2011), including studies which have used the same sediment cores as used here. We do not want to repeat their conclusions in our manuscript. Rather, we explicitly chose these cores and regions as much is already known about the paleoclimate, which makes it easier to understand the behavior of a new proxy. What is known about the past climate in these regions is now summarized in the method section: lines 126-132 for the Zambezi River: "To summarize, during H1 and the YD, the Zambezi catchment is characterized by higher precipitation and enhanced riverine runoff due to a southward shift of the Intertropical Convergence Zone (ITZC) resulting from Northern Hemisphere cold events, whereas during the Holocene drier conditions prevailed (Schefuß et al., 2011; Wang et al., 2013; van der Lubbe et al., 2014; Weldeab et al., 2014). The Last Glacial Maximum (LGM) in the Zambezi catchment is also recognized as an extremely wet period (Wang et al., 2013)." and lines 158-163 for the Nile River: "To summarize, the climate of the Nile catchment area was colder and drier (Castañeda et al., 2010, 2016) during the YD, H1 and the LGM. The LGM and H1 were extremely arid events with the likely desiccation of the Nile water sources, i.e. Lake Tana and Lake Victoria (Castañeda et al., 2016). To the contrary, the time period during S1 sapropel deposition was warmer and wetter resulting in an enhanced riverine runoff. The late Holocene is characterized by a decrease in precipitation (Blanchet et al., 2014)."**

*-The reviewer indicates that part of the difference in correlation between the C$_{32}$ 1,15-diol and BIT between the sites can be due to the different hydrological setting of the rivers.*

**We find it difficult to explain the difference in correlation between F$_{C32\ 1,15}$ and the BIT index by different hydrological settings and/or the length of the river alone. Furthermore, as noted in the text, these two records generally agree well with each other.**

*-The reviewer mentions that the δD decreases between 35 and 38ky (line 319) and does not increase leading to drier conditions and not more humid as stated in our manuscript.*
**The reviewer is correct, and we will delete this part of the discussion as the change in sea level is enough to explain the increase in input of $C_{32}$ 1,15-diol and brGDGTs in our core.**

*-The reviewer indicates that we use the same environmental factor, i.e. aridity, to explain two different observations, i.e. a decrease in riverine input and an increase in soil erosion (line 383).*
**Data supporting the change in vegetation and aridity during 0-5ky have been reported by Blanchet et al. (2014) and this is mentioned in the manuscript at line 429, while data supporting the extreme aridity during H1, more than during 0-5 ky, have been reported by Castaneda et al. (2016). It might be that the extreme aridity during H1 (when both Lake Tana and Lake Victoria, the sources of the Blue and White Nile, were desiccated) led to a lack of vegetation and increased soil erosion but it also substantially reduced river flow, such that net export of soil OM was reduced. In contrast for the period between 0-5 ky, aridity was not as severe and thus the increased soil erosion combined with a moderately reduced river flow still leads to export of terrigenous OM as also shown by the relatively higher Ca/Ti (Castaneda et al., 2016). We have expanded the explanation at lines 401-404: ". These low values can be attributed to extreme aridity in the Nile River catchment (Castañeda et al., 2016) which we hypothesized lead to a lack of vegetation and enhanced soil erosion but also leading to a severely reduced low river flow, thereby decreasing the net amount of river borne OM reaching our core site."**

*-The reviewer states that a decrease in log(Ca/Ti) during 0-5 ka indicates more riverine input.*
**It is true that log(Ca/Ti) is decreasing during 0-5ka and that it indicates more soil input, but it does not indicate per se more riverine input itself. We have added more detail concerning our idea at line 443-444: "This hypothesis is supported by the log (Ca/Ti) (Fig. 4c) which is decreasing at this time, suggesting that soil run off was increasing."**

*- The reviewer noticed that at line 421 the figure name should be S3b and not Fig. 3.*
**This has been corrected.**

*-The reviewer is asking why the axis for log(Ca/Ti) has been inverted.*
**In our view, by flipping the log(Ca/Ti) we highlight the similarity between this proxy and the BIT and $C_{32}$ 1,15-diol, making the correspondence between the three proxies clearer.**

*- The reviewer says that in fig. 4 the BIT index and log(Ca/Ti) are more similar than the %$C_{32}$ 1,15 and log (Ca/Ti).*
**The BIT index follows the log(Ca/Ti) better than the diol index as they both are influenced by riverine input of soil-derived carbon while the diol index is not tracing this pool but rather river-borne carbon.**

*-The reviewer is asking if the concentration of the 1,13 and 1,14 diols are available.*

**Unfortunately, it was not possible to quantify the concentrations of the 1,13 and 1,15 diols as they were measured from long term stored archived extracts.**

**Reviewer #2:**

We thank the reviewer for his/her helpful comments on our manuscript. Below follows our reply to the main comments and, if applicable, how we changed the manuscript.

*-The reviewer wishes to have more information on LCDs (potential sources, history).*
**We have extended the third paragraph of the introduction with more details about the discovery of LCDs and potential producers in marine and freshwater environments (lines 69-77): "Long-chain diols (LCD), such as the $C_{32}$ 1,15-diols, are molecules composed of a long alkyl chain ranging from 26 to 34 carbon atoms, an alcohol group at position $C_1$ and mid-chain position, mainly at positions 13, 14 and 15. They occur ubiquitously in marine environments (de Leeuw et al., 1979; Versteegh et al., 1997, 2000; Gogou and Stephanou, 2004; Rampen et al., 2012, 2014; Romero-Viana et al., 2012; Plancq et al., 2015; Zhang et al., 2011 and references therein), where the major diols are generally the $C_{30}$ 1,15-, $C_{28}$ and $C_{30}$ 1,13-ls and the $C_{28}$ and $C_{30}$ 1,14-diols. In marine environments the 1,14-diols are produced mainly by Proboscia diatoms (Sinninghe Damsté et al., 2003, Rampen et al., 2007) and the 1,13 and 1,15-diol are thought to be produced by eustigmatophyte algae"**

*-The reviewer would like to have the age control points indicated in figure 2 and 4 and a supplement with a summary of the age model.*
**The age control points have been added to the figures (black triangles) and a brief supplementary method section describing the previously published age-models of both cores is also added.**

*-The reviewer wishes to have a more synthetic discussion on the proxy, with more details about its advantages and disadvantages, as well as why, in the Mozambique core, the correlation between the $C_{32}$ 1,15-diol and BIT index is better than in the Nile core but also why the $C_{32}$ 1,15-diol works better than the BIT.*
**The $C_{32}$ 1,15-diol is not working 'better' than the BIT index to trace riverine input, rather in our view it simply reflects a different pool of organic carbon being transported by rivers, i.e. river-born carbon versus soil and river-born carbon in case of branched GDGTs. This is now better detailed at lines 5, 7, 93. Like the BIT index, the $F_{1,15-C32}$ may also be affected by marine productivity as we discussed at lines 352-355, 414-417and 419-423. For the different observations between the cores, we added some speculation at lines 446-448: "Although some discrepancies are noted for both cores, the $C_{32}$ 1,15-diol agrees well with other terrigenous proxies." A synthesis of the advantage/disadvantages of the proxy are discussed at lines 451-454: "Since the $C_{32}$ 1,15-diol is produced in rivers itself, it is not impacted by vegetation abundance and soil composition, in contrast to other proxies like the BIT index and lignin concentrations. This may make it a more reliable proxy to trace past river input into marine environments."**

*-The reviewer asks for a clearer discussion on the source of the C$_{32}$ 1,15-diol and what does the synchronicity/asynchronicity of the variation between BIT index and C$_{32}$ 1,15-diol means in a broader sense.*
**In our view, this question has been discussed already at lines 345-364 and 405-417 and we want to point out that for most of the records, BIT index and C$_{32}$ 1,15-diol actually agree quite well. Furthermore, in essence, the proxies record different things so they are not always behaving the same way. This is now better explained.**

*-The reviewer indicates that the hypothesis at lines 366-371 is more speculative than other part and that, if true, the low brGDGT concentration of the Northern Rivers would be reflected in surface sediment offshore these rivers.*
**We agree with the reviewer that this is speculative, and therefore we added at line 382: "We hypothesize that.." and at lines 387-390, we recommend for future studies to analyze surface sediments offshore the Northern rivers to confirm this hypothesis: " Further research examining the brGDGT contents of soils in the different river catchment areas as well as surface sediment from offshore these northern rivers is required to distinguish between the different hypotheses."**

*- The reviewer would like, if possible, to have more general discussion on the paleoclimate during H1 and YD.*
**We have added some details about the climate during the YD and H1 in the method section at lines 127-132 for the Zambezi River: "To summarize, during H1 and the YD, the Zambezi catchment is characterized by higher precipitation and enhanced riverine runoff due to a southward shift of the Intertropical Convergence Zone (ITZC) resulting from Northern Hemisphere cold events, whereas during the Holocene drier conditions prevailed (Schefuß et al., 2011; Wang et al., 2013; van der Lubbe et al., 2014; Weldeab et al., 2014). The Last Glacial Maximum (LGM) in the Zambezi catchment is also recognized as an extremely wet period (Wang et al., 2013)." and lines 158-163 for the Nile River: "To summarize, the climate of the Nile catchment area was colder and drier (Castañeda et al., 2010, 2016) during the YD, H1 and the LGM. The LGM and H1 were extremely arid events with the likely desiccation of the Nile water sources, i.e. Lake Tana and Lake Victoria (Castañeda et al., 2016). To the contrary, the time period during S1 sapropel deposition was warmer and wetter resulting in an enhanced riverine runoff. The late Holocene is characterized by a decrease in precipitation (Blanchet et al., 2014)."**